# The Broad-Spectrum Endolysin LySP2 Improves Chick Survival after *Salmonella* Pullorum Infection

**DOI:** 10.3390/v15040836

**Published:** 2023-03-24

**Authors:** Hewen Deng, Mengjiao Li, Qiuyang Zhang, Chencheng Gao, Zhanyun Song, Chunhua Chen, Zhuo Wang, Xin Feng

**Affiliations:** 1College of Veterinary Medicine, Jilin University, Xi’an Street 5333#, Changchun 130062, China; 2Changchun Customs District, Changchun 130000, China

**Keywords:** *Salmonella* pullorum, endolysin, *Pichia pastoris*, expression

## Abstract

*Salmonella* pullorum causes typical “Bacillary White Diarrhea” and loss of appetite in chicks, which leads to the death of chicks in severe cases; thus, it is still a critical issue in China. Antibiotics are conventional medicines used for *Salmonella* infections; however, due to the extensive long-term use and even abuse of antibiotics, drug resistance becomes increasingly severe, making treating pullorum disease more difficult. Most of the endolysins are hydrolytic enzymes produced by bacteriophages to cleave the host’s cell wall during the final stage of the lytic cycle. A virulent bacteriophage, YSP2, of *Salmonella* was isolated in a previous study. A *Pichia pastoris* expression strain that can express the *Salmonella* bacteriophage endolysin was constructed efficiently, and the Gram-negative bacteriophage endolysin, LySP2, was obtained in this study. Compared with the parental phage YSP2, which can only lyse *Salmonella*, LySP2 can lyse *Salmonella* and *Escherichia*. The survival rate of *Salmonella*-infected chicks treated with LySP2 can reach up to 70% and reduce *Salmonella* abundance in the liver and intestine. The treatment group showed that LySP2 significantly improved the health of infected chicks and alleviated organ damage caused by *Salmonella* infection. In this study, the *Salmonella* bacteriophage endolysin was expressed efficiently by *Pichia pastoris*, and the endolysin LySP2 showed good potential for the treatment of pullorum disease caused by *Salmonella* pullorum.

## 1. Introduction 

At the beginning of the last century, *Salmonella* pullorum and *Salmonella* Typhoid were two important pathogens that caused pandemics in chicken flocks in Europe and the United States [1]. Especially in chicks within four weeks of age, the morbidity and mortality rates are incredibly high, seriously affecting breeders’ hatchability and survival rate and bringing substantial economic losses to the poultry industry. Eliminating *Salmonella* pullorum from chicken farms is the best way to eradicate Pullorosis disease [2]. Compared with antibiotics’ shortcomings, phages and endolysin have great potential as new antibacterial agents [3,4]. Phage endolysins, which have high lytic efficiency, are associated with a low probability of bacterial resistance, and can be combined with antibiotics [5]. These advantages improve prospects for developing new drugs [6,7]. Due to the bacterial outer membrane, it is difficult for the endolysin of a Gram-negative bacteriophage to come into contact directly with the reaction substrate, resulting in low lytic activity during direct use [8,9]. Endolysins can be used in combination with polycations, weak organic acids, chelators, and other agents that can increase outer membrane (OM) permeability [10,11], enabling endolysins to enter and hydrolyze the peptidoglycan layer, causing the host cell to rupture due to increased internal osmotic pressure. In addition, endolysin is combined with holin [12,13,14] to enhance lysis activity. These provide novel approaches for optimizing endolysin efficacy. Additionally, there are endolysins that, through lytic transglucosylases (e.g., phage lambda endolysin), perform non-hydrolytic cleavage of the glycan strand of bacterial murein with the formation of the non-reducing *N*-acetyl 1,6-anhydromuramic acid, and *N*-acetylglucosamine [15,16]. Most research on phage endolysins has emphasized its activity against Gram-positive bacteria [17,18]. However, a few studies focused on using Gram-negative phage endolysins directly [19]. In these studies, the endolysin expressed by *Pichia pastoris* showed high lytic activity when used directly. The experimental design of this work can be roughly described as follows: the natural endolysin with antibacterial activity was produced using a eukaryotic expression system, the endolysin was then secreted and expressed in the culture supernatant, then its biological properties and antibacterial tests were characterized in vitro, then direct oral use in animal experiments was carried out to verify the therapeutic effect of endolysin on infected animals. The aim of this work was to develop endolysins as a new alternative antibacterial substance to eliminate pathogenic bacteria and lay the foundation for phage endolysins’ wide and direct use.

## 2. Materials and Methods

### 2.1. Phage, Bacterial Strains, and Culture Conditions

*Salmonella* pullorum was isolated from a poultry farm with diseased chickens in Changchun, China, and stored in the laboratory [20]. Using this bacterium as a host, phage YSP2 (NC_047898) was isolated from various sewage systems in Changchun, China, and stored in the laboratory [20]. *Salmonella* pullorum was cultured in 5 mL Luria–Bertani (LB) broth at 37 °C, 160 rpm. After growing to an optical density at 600 nm = 0.6, 100 μL of YSP2 (10^9^ PFU/mL) was added, and the culture was shaken at 37 °C, 160 rpm until the medium was clarified. The lysate was then filtered through 0.22-μm microporous membranes to obtain the phage YSP2 suspension. *Pichia pastoris* X33 was maintained in the laboratory [21].

### 2.2. Amino Acid Sequence Analysis of LySP2

The amino acid sequence of LySP2 (YP_009795998) was aligned using BLAST (https://blast.ncbi.nlm.nih.gov, accessed on 8 March 2020.), and several phylogenetically-related protein amino acid sequences were used to reconstruct a phylogenetic tree using MEGA7 software [22]. The amino acid sequence was aligned using the DNAMAN software.

### 2.3. Construction of Expression Vector pPICZ-LySP2

A 495-bp DNA fragment containing the *lysp2* gene was amplified by a polymerase chain reaction (PCR) using phage YSP2 genomic DNA as template and primers LySP2-ZF (CCGCTCGAGAAAAGAATGGCTATTAAAAAGACAATAGCCG) and LySP2-ZR (TGCTCTAGATCATTTATTCAGATCCATTACACAA). Restriction endonuclease sites for *Xho*I and *Xba*I were included at the 5′ ends of LySP2-ZF and LySP2-ZR, respectively. Primers were synthesized by Shanghai Shengong Bioengineering Co., Ltd., Shanghai, China. The PCR product was digested with *Xho*I (Takara, Shiga, Japan) and *Xba*I (Takara) and ligated into the corresponding sites of vector pPICZ-αA (Invitrogen, Carlsbad, CA, USA) using T4 DNA ligase (Takara) to construct pPICZ-LySP2. The ligation product was transformed into *E. coli* strain DH5α, and positive transformants were selected using the Zeocin+ (25 μg/mL) low salt LB solid medium [21], from which plasmids were extracted for PCR and sequencing.

### 2.4. Construction of Recombinant P. pastoris Strain X33-pPICZ-LySP2

Following sequence confirmation, the plasmid pPICZ-LySP2 and the empty vector pPICZ-αA were linearized using the restriction endonuclease *Sca*I (Takara). Digested products were electro-transformed into *P. pastoris* X33. Ten micrograms of linearized plasmid were mixed with 80 μL of competent yeast cells on ice, gently mixed, and transferred to a 2-mm electroporation cuvette. The following electroporation conditions were used: 1500 V, 200 Ω, and 25 μF. After transformation, 1 mL of pre-cooled 1 M sorbitol was added to the cuvette, mixed, and transferred to 2 mL of the Yeast Extract Peptone Dextrose (YPD) liquid medium. After brief incubation at 30 °C for 30 min, 100 μL of the cell mixture was plated on YPD Zeocin+ (100 μg/mL) agar plates and cultured at 29 °C for two days until colonies grew. Single colonies were subsequently cultured in a YPD medium containing Zeocin+ (100 μg/mL) at 29 °C for 12 h, after which DNA was extracted using a yeast genome extraction kit (TIANGEN, Beijing, China). PCR was performed to identify a positive recombinant yeast strain and an empty vector yeast strain, named X33–pPICZ–LySP2 and X33-pPICZ, respectively.

### 2.5. Induced Fermentation and Optimization of Recombinant P. pastoris X33–pPICZ–LySP2

Positive recombinant yeast strains were inoculated into 25 mL of Buffered Minimal Glycerol (BMGY) liquid medium and grown at 29 °C, 350 rpm, to an OD_600_ of 2.0. The culture was centrifuged at 2500× *g* for 5 min at 4 °C. Sterile phosphate-buffered saline (PBS) was used to wash the cell pellet twice to remove glycerol completely. Cells were then transferred to 50 mL of Buffered Minimal Methanol (BMMY) medium at an initial OD_600_ of 1.0 and grown at 29 °C, shaking at 350 rpm for 72–96 h. Methanol (1%, *v*/*v*) was added to the medium every 12 h to induce strain X33–pPICZ–LySP2 to produce endolysin (details of the optimal expression conditions are shown in Appendix A). Samples were analyzed for cell growth every 24 h to screen for strains with high endolysin production. The size of LySP2 was detected using sodium dodecyl sulfate–polyacrylamide gel electrophoresis (SDS-PAGE) analysis (details of the optimal eukaryotic expression conditions of LySP2 are shown in Appendix A).

Fermentation supernatant from strains with high endolysin production was added to a 50-kDa molecular weight cutoff ultrafiltration device, centrifuged at 2000× *g* for 15 min, and the filtrate in the centrifuge tube was collected. The process was repeated with a 3-kDa molecular weight cutoff ultrafiltration device with centrifugation at 2000× *g* for 15 min. The filtrate was concentrated until a volume of less than 1 mL was reached and then saved for later use. The concentrated sample was named LySP2; the same process was used for the fermentation supernatant from strain X33–pPICZ, and the sample was called pPICZ. *Salmonella* pullorum suspensions (100 μL) at a concentration of about 10^8^ colony-forming units (CFU)/mL were used to coat the surface of an LB plate evenly. A 6-mm-diameter hole was punched into the agar, and 100 μL of the LySP2 was added, with pPICZ added to the second hole as a control. The plate was incubated at 37 °C for 24 h, after which the bacteriostatic diameter was recorded and averaged (*n* = 3).

### 2.6. Determination of the Endolysin Range of LySP2

The agar diffusion method was used to detect antibacterial activity against the following test bacteria (Table 1). Bacterial suspensions (100 μL) at a concentration of about 10^8^ CFU/mL were used to coat an LB plate evenly. Holes were punched as described above, and 100 μL of the LySP2 was added. Plates were incubated at 37 °C for 24 h, after which the bacteriostatic diameter was recorded and averaged (*n* = 3).

### 2.7. Determination of Minimum Inhibitory Concentration (MIC) of LySP2

*Salmonella* pullorum was collected during the exponential growth phase (around 10^6^ CFU/mL) and diluted at 1:100. LySP2 at 100 μL was diluted with PBS (pH 7.0) to 60, 30, 15, 7.5, 3.75, 1.88, 0.94, 0.47, 0.23, and 0.12 µg/mL. For control, diluted *Salmonella* pullorum was treated separately with penicillin or the pPICZ. After incubation for 12–24 h at 37 °C, the OD_600_ value of the culture medium was measured using a multi-functional enzyme label instrument (BioTek, Elx800, Winooski, VT, USA). The minimum LySP2 concentration that could completely inhibit bacteria growth was determined as the MIC (*n* = 3).

### 2.8. Stability Test of LySP2

LySP2 stability was assessed by measuring the inhibition zone’s diameter after treatment under different pH and temperature conditions. For pH stability tests, the LySP2 was incubated at room temperature (pH 3–9) for 1 h. For thermal stability tests, the LySP2 was set at different temperatures (4, 30, 40, 50, 60, 70, 80, 90, and 120 °C) for 30 min, cooled to room temperature, and a 100 μL sample was taken to measure the diameter of the inhibition zone using the agar diffusion method. The diameter of the inhibition zone produced by the untreated LySP2 was used as a control to compare the percentage of lytic activity after each treatment.

### 2.9. LySP2 Treatment of the Salmonella Pullorum Chick Infection Model

Chicks (3-day-old, male and female) hatched from SPF eggs were used in this study for the infection model. We constructed the *Salmonella* pullorum-infected chick model according to L. Revolledo’s [23] method and improved it. Five groups (*n* = 10 each) of 3-day-old chicks were fasted for 12 h and received different oral treatments (Table 1). Chicks were observed for 14 days to determine the survival rate.

After 7 days of treatment, the health status of chicks in each group was observed and scored, and the scoring method was modified from that of S. Shivaramaiah [24]. Scoring criteria were as follows: 0 points: death; 1 point: crouching on the ground, not eating, dying; 2 points: closed eyes, abnormal posture, anus closed by excrement; 3 points: lack of energy, reduced diet, dysplasia, perianal villi stained by feces; 4 points: cold sensitivity, activity reduced; 5 points: normal health, activity, and eating patterns. After scoring, all chicks were euthanized by cervical displacement for harmless treatment of chick carcasses, according to the Animal Welfare and Research Ethics Committee requirements at Jilin University.

After 7 days of treatment, blood from each group of chicks was collected, mixed with anticoagulant, and centrifuged at 2000× *g* for 2 min at 4 °C. The endotoxin content in chicken serum was determined using a Pierce™ Chromogenic Endotoxin Quant Kit (Thermo Fisher Scientific, Waltham, MA, USA), reading absorbance at 405 nm.

To determine changes in body temperature, the temperature under the wings was measured every day using an electric thermometer.

### 2.10. Counts of Salmonella in the Liver and Intestine

Fifty chicks were challenged and treated as described in Table 1. After 3, 5, and 7 days, three chicks were randomly selected from each group and sacrificed by cervical dislocation. The liver and intestines of the chicks were aseptically harvested, weighed, homogenized, and ground in PBS. Liver and intestinal sera were diluted 10 times in PBS, and 100 μL of each dilution was spread on Salmonella–Shigella agar (Biokar Diagnostics, Allonne, France) and cultured in a 37 °C incubator for 24 h, after which the number of bacteria per gram of tissue was determined.

### 2.11. Pathological Observation of Chick Intestinal and Liver Tissues

After treatment for 7 days (Table 2), chicks were sacrificed by cervical dislocation. The liver and intestines were removed, and gross pathological changes in the liver and intestinal surfaces were observed. Tissue samples were fixed in buffered formalin solution (4%). Sections were stained with hematoxylin and eosin (H&E), and pathological changes were observed under a microscope. For villus height/crypt depth ratio determination, a quantitative analysis of intestinal pathological tissue sections was performed using a Motic (Hong Kong, China) digital fiber image processing system, measuring villus height (from the top of the villus to the crypt opening) and the depth of the crypt connected to the villi (from the bottom of the nest to the level of the crypt opening). The ratio of the height of the villi to the depth of the crypt was calculated by dividing the height of the villi by the depth of the crypt.

### 2.12. Bioethics

All animal management and experiments were strictly based on the Regulations for the Administration of Affairs Concerning Experimental Animals approved by the State Council of the People’s Republic of China (3.1.2017) and approved by the Animal Welfare and Research Ethics Committee at Jilin University (SY201902009).

### 2.13. Statistical Analysis

The data are expressed as mean ± SD. All experiments were independently performed three times. *t*-tests were applied to assess the difference when two groups were compared (where *p* < 0.05 indicates significant differences and *p* < 0.01 shows extremely significant differences). A Tukey test was applied when examining changes in endotoxin and body temperature in different groups (where *p* < 0.05 indicates significant differences). Statistical analysis was performed using GraphPad Software (GraphPad Software Inc., San Diego, CA, USA).

## 3. Results

### 3.1. Comparative Analysis of LySP2 with Another Phage

Endolysin LySP2 contains 164 amino acids, and BLAST analysis showed that its amino acid sequence is highly similar to the amino acids of endolysin from other *Salmonella* bacteriophages. Notably, the percentage identification of LySP2 with endolysin of *E. coli* bacteriophage TLS (YP 001285558.1), *E. coli* phage ev116 (VUF54951.1), and *Citrobacter* phage Stevie (YP 009148745.1) reached 95.73%, 93.90%, and 92.68%, respectively (Appendix A). The amino acid sequence used for the BLAST analysis was employed for phylogenetic tree construction, which showed that endolysin amino acids from other phages are closely related to LySP2 (Figure 1). Analyzing the amino acid sequences of LySP2 and those from *E. coli* phage TLS and *Citrobacter* phage Stevie by DNAMAN, the similarity was 97.56% (Appendix A). Through comparative analysis and alignment of phylogenetic relationships, we suggest that LySP2 is a phage endolysin.

### 3.2. Construction and Expression of LySP2

We extracted the whole genome of YSP2 (GenBank: MG241338.1). The target gene *lysp2* was amplified by PCR and used to construct the recombinant plasmid pPICZαA–LySP2. After verification by double enzyme digestion (Figure 2A) and PCR (Figure 2B), the recombinant plasmid pPICZαA–LySP2 was transformed into competent *P. pastoris* X33 cells to obtain the recombinant yeast expression strain X33–pPICZ–LySP2. The molecular weight of the LySP2 protein expressed by recombinant yeast X33–pPICZ–LySP2 was 18.2 kDa. (Figure 2C), and the concentration was 238.76 µg/mL (details of the optimal eukaryotic expression conditions of LySP2 are shown in Appendix A).

### 3.3. General Biological Characteristics of LySP2

We determined the lytic activity of LySP2 against *Salmonella* pullorum using the agar diffusion method. A larger inhibition zone indicated that LySP2 exhibited stronger lytic activity against Salmonella pullorum than the control (Figure 3A).

Next, we explored the stability of LySP2. Following treatment at 0–120 °C for 30 min, LySP2 maintained lytic activity between 0 and 40 °C, but it gradually decreased with an increase in temperature (Figure 3B). LySP2 retained more than 85% activity after treatment at pH 3–9 (Figure 3C), indicating that it has a broad pH tolerance range.

It is well known that phages are extremely specific to the host. The parental phage YSP2 has lytic activity against *Salmonella* strains but no lytic activity against other genera under laboratory conditions [20]. Therefore, we examined the lytic spectrum of LySP2 and found that it retains lytic specificity for *Salmonella* and *E. coli* (Figure 3D, Details of the antibacterial diameter shown in Appendix A).

The MIC of LySP2 was also determined. The concentration of bacteria was significantly reduced at a LySP2 concentration as low as 3.75 μg/mL (Figure 3E).

### 3.4. Therapeutic Effect of LySP2 in the Salmonella Pullorum-Infected Chick

We successfully constructed a *Salmonella* pullorum-infected chick model. After LySP2 treatment, the chick survival rate was 70% of that of the uninfected normal group and slightly higher than the survival rate (60%) of the phage group (*p* > 0.05, Figure 4A), indicating that LySP2 can improve the survival rate of infected chicks.

The health status of chicks in the LySP2 and phage groups was significantly better than that of the pPICZ and control groups (Figure 4B). This indicated that LySP2 could improve the health of infected chicks.

Endotoxin test results are shown in Table 3. The endotoxin concentration in the blood of chicks in the control group was the highest, but there was no significant difference among the control group, pPICZ group, and LySP2 group (*p* > 0.05). The endotoxin concentration in the phage group was significantly lower than that in the LySP2 group (*p* < 0.05).

The body temperature of healthy chicks under 10 days old is about 38 °C. The body temperature of 5-day-old chicks was not significantly different among treatment groups. However, the body temperatures of chicks in the LySP2, control, and pPICZ groups were significantly different from that of the normal group at 7 days of age (*p* < 0.05, Table 4). The body temperature of 9-day-old chicks in the normal group significantly differed from all other groups (*p* < 0.05). This indicates that LySP2 can stabilize body temperature and improve health in infected chicks.

After 3, 5, and 7 days of treatment, we assessed *Salmonella* abundance in the livers (Figure 5A) and intestines (Figure 5B). There were significant differences in *Salmonella* abundance between chicks in the phage or LySP2 groups and those in the pPICZ group (*p* < 0.01). The clearance rates of *Salmonella* in the livers of chicks in the phage and LySP2 groups were 83.33% and 100%, respectively. Intestinal cell counts revealed 50% *Salmonella* clearance rates for chicks in both the phage and LySP2 groups. Intestines of chicks in the control and pPICZ groups contained 10^8^ CFU *Salmonella* per mg tissue. LySP2 reduced Salmonella abundance in the intestines of infected chicks, and Salmonella infection was insufficient to spread to the liver.

After 7 days of treatment, infected chicks were dissected to obtain liver and duodenum tissue for sectioning. We also conducted a gross inspection of livers. Compared with livers from the normal group (Figure 6E), which were yellow and soft in texture, livers from the control group (Figure 6A) and pPICZ group (Figure 6B) were congested and swollen, dark red, soft, and brittle in texture. There were grayish-yellow necrotic lesions and cheese-like stripes on the surface. Livers in the phage group (Figure 6C) and LySP2 group (Figure 6D) were a normal color, but a small amount of bleeding was observed.

H&E staining revealed congestion and hemorrhage in liver capillaries of the control group (Figure 6A) and the pPICZ group (Figure 6B). The hepatic cord spacing was widened, indicative of edema. Necrotic foci of different sizes appeared in the hepatic lobules, and numerous inflammatory cell infiltrations were found in necrotic foci. Livers of chicks in the phage group (Figure 6C) and LySP2 group (Figure 6D) showed significantly fewer pathological changes, but the tissue was slightly edematous.

After 7 days of treatment, the intestinal tissue of chicks in the control group (Figure 7A) and pPICZ group (Figure 7B) showed broken muscle, congestion, wave-like degeneration, necrosis, and a thickened serosa layer. Large numbers of inflammatory cells had infiltrated, and duodenal villus height was lower than that of the normal group (*p* < 0.01). Intestinal tissue in the phage group (Figure 7C) and LySP2 group (Figure 7D) displayed a normal color, slight congestion in the intestinal mucosa, and a small number of lymphocytes infiltrated into the lamina propria. The ratio of villus height to crypt depth was significantly higher than that of the control group (*p* < 0.05). The normal group (Figure 7E) showed no apparent abnormalities. These results indicated that LySP2 could ameliorate pathological changes in infected organs and repair tissue damage caused by *Salmonella* infection.

## 4. Discussion

Our phage YSP2 sequencing results showed that the phage has a holin-like gene [20]. Most phages’ lytic effects result from synergy between endolysin and holin [13,25,26,27]. Holins are small membrane proteins that accumulate in the membrane until, at a specific time that is “programmed” into the holin gene, the membrane suddenly becomes permeable to the fully folded endolysin. Destruction of the murein and bursting of the cell are immediate sequelae [26]. Our previous research revealed that phage holin proteins show a broad antibacterial range [28], and combinations of endolysins and holin may be potential novel antibiotic candidates [25,29]. The mechanism of synergistic action between holin and the LySP2 endolysin remains to be elucidated. Most research has focused on exploring gene sequences and evolutionary relationships between pathogenic bacteriophages and their ability to lyse pathogens [30]. Many purified phage endolysins have shown antibacterial activities in microbial culture experiments. LysAP contains a C-terminal region transmembrane domain, which is crucial for protein localization and bacterial lysis [27]. The intrinsically disordered and highly positively charged N-terminal region of the enzyme of LysC can form nanopores in the cytoplasmic membrane and can kill bacteria by destabilizing the bacterial cell membrane through electrostatic interactions. this is another mechanism by which endolysin lyses bacteria [31]. Natural endolysins have been targeted for truncations or manipulation to explore their bactericidal mechanisms and broaden the antibacterial spectrum [32]. We confirmed the in vitro lytic activity of LySP2 and treated infected chicks. The results demonstrated that LySP2 could successfully treat *Salmonella* pullorum. After LySP2 is expressed, without any pretreatment, it shows an excellent lytic effect. In future applications, we will consider using it with additional reducing agents to explore if lysis potential can be increased.

We detected the antibacterial activity of phage endolysin LySP2 against 11 strains of bacteria. LySP2 had a broader lytic spectrum than phage YSP2, possessing lytic activity against *Salmonella* and *Escherichia* strains. The whole-genome sequence of *Salmonella* phage YSP2 was examined [20], and the phage endolysin LySP2 showed high sequence similarity with some *E. coli* phage endolysins. We, therefore, speculated that the lytic enzyme LySP2 might have lytic activity against bacteria such as *E. coli*. The results in Figure 1 show that LySP2 is similar to the endolysin of *salmonella* and the endolysin of two strains of *E. coli* phage (TLS, Stevie), and one *Citrobacter* phage also has a high affinity with LySP2. This is one of the reasons why LySP2 can lyse *E. coli*. A previous study reported that some endolysins’ lytic spectra are beyond the host’s scope [33]. For example, the *A. baumannii* phage endolysin inhibits *A. baumannii* and other strains, such as *Escherichia coli* and methicillin-resistant *Staphylococcus aureus* (MRSA) [34]. *Pseudomonas* bacteriophage ZCPS1 endolysin inhibits Gram-positive and Gram-negative bacteria [35]. Endolysin primarily lyse host bacteria by hydrolyzing peptidoglycans in the cell wall, disrupting its structure [36]. Certain peptide chains, such as glycosidic linkages of acetylglucosamine, are highly conserved among bacterial species so that endolysin may be linked to the conserved structure of cell wall peptidoglycans [37,38]. This may be why endolysins have a broader lytic spectrum than phages.

The widely used *P. pastoris* expression system has high biosafety and low toxicity, and the structures and activities of expressed proteins are close to those of the natural proteins [39,40,41]. There is no non-specific activation or inhibition of target DNA or eukaryotic genes, since the target DNA has no homology with the eukaryotic gene regulatory sequence [42]. The eukaryotic expression system can regulate gene secretion and overcome challenges in purifying prokaryotic expression system products [43]. In the present study, a prokaryotic expression system was used to express the *Salmonella* pullorum phage endolysin LySP2. We unexpectedly found that after transferring the prokaryotic recombinant plasmid into the expression strain *E. coli* BL21 and inducing protein expression, the bacterial suspension began to break at an OD_600_ of about 0.4–0.5, and the bacterial liquid rapidly became clear. The strain with induced LySP2 endolysin production could not continue to grow, and we could not obtain the expression product of LySP2 (Appendix A). This demonstrated that LySP2 had lytic activity on *E. coli*. We, therefore, used a eukaryotic expression system that is more conducive to late expression, purification, and production of the endolysin [44] by exocrine secretion.

Phage endolysins have wide temperature and pH tolerance ranges and maintain activity following short-term changes in these factors. For example, the endolysin Ply 5218 can maintain activity at 50 °C for more than 30 min, and its activity still exceeds 70% at 70 °C [45]. The lytic activity of endolysin P28 is maintained between pH 4 and 12 [46]. In this study, LySP2 lytic activity was retained after exposure to temperatures of 4–40 °C for 30 min. At 50–90 °C, lytic activity showed a significant decrease to 54.17% of that of controls. The lytic activity of lysin Ply5218 was only 5.56% after treatment at 120 °C for 30 min [45]. Moreover, endolysin LySP2 tolerated a pH between 3 and 9, with activity maintained at >85%. These results are consistent with those of a previous study [46]. Endolysin activity can also be affected by enzyme concentration, metal ions, and salt concentration, and various factors have been shown to have similar effects on different endolysins [47]. Therefore, the therapeutic effect of the LySP2 endolysin may be improved by adding additional cofactors.

Hepatitis is known to cause an increase in blood endotoxin levels due to liver dysfunction, which impairs the body’s ability to detoxify endotoxins [45,46]. However, treatment with endolysin LySP2 has been found to reduce endotoxin levels in chicks infected with Salmonella. It is believed that this is because LySP2 reduces the abundance of pathogenic bacteria in the early stages of infection, allowing the body’s clearance mechanisms to eliminate endotoxins more effectively. In addition to reducing endotoxin levels, LySP2 also appears to alleviate organ pathology in infected chicks. This is evidenced by the reduced inflammatory cell infiltration, necrosis, and hepatic cord gap observed in liver tissue samples from the LySP2 group compared to the control group. Moreover, the ratio of duodenal villus height to crypt depth was higher in the LySP2 group, indicating that LySP2 treatment also helped to maintain normal physiological function in the intestine. Pathological changes in the intestinal epithelium can lead to decreased nutrient absorption, decreased disease resistance, and reduced production performance [47]. However, LySP2 treatment was found to reduce these negative effects by reducing the depth of the crypt and increasing the height of the villi, thereby reducing turnover of the epithelium and lowering the demand for nutrients. Overall, these findings suggest that treatment with endolysin LySP2 may be an effective strategy for reducing the negative effects of pathogenic bacteria on the liver and intestine. By reducing the abundance of pathogenic bacteria in the early stages of infection, LySP2 treatment may help to alleviate organ pathology and maintain normal physiological function in infected individuals.

*Salmonella* pullorum often causes acute systemic disease in chicks, and a considerable effort has been made to control and prevent it in the poultry industry. In countries where prevention and control measures are inefficient and climate conditions are conducive to spreading *Salmonella*, the disease continues to have severe economic impacts on poultry farming. *Salmonella* pullorum strains isolated from diseased chickens in China from 1962 to 2007 were reported to have high levels of resistance to ampicillin, streptomycin, tetracycline, and trimethoprim [48], and the problem of multi-drug resistance is getting worse. There is considerable interest in exploring and developing new drugs to prevent *Salmonella* pullorum infection that can replace traditional antibiotics without drug resistance. Guo et al. developed an attenuated strain combined with protein E-mediated cell lysis to produce *Salmonella* pullorum ghost vaccines, which increased lymphocyte proliferation responses to protect *Salmonella*-infected chicks [49]. Others have tested the effects of Chinese herbal medicines [50], acidifiers, oligosaccharide additives, and enzyme preparations. Zhao and colleagues [51] isolated a virulent bacteriophage that is another promising tool against *Salmonella* infection. Compared with phages, recombinant endolysins have more potent effects on host bacteria, a broad lytic spectrum, and great potential for increased yield through genetic engineering. Phage endolysins have also shown the advantages of high efficiency, safety, and minimal antibiotic resistance in the skin, nasal mucosa, vein, endocarditis, endophthalmitis, meningitis, and pneumonia infection models [52,53].

## Figures and Tables

**Figure 1 viruses-15-00836-f001:**
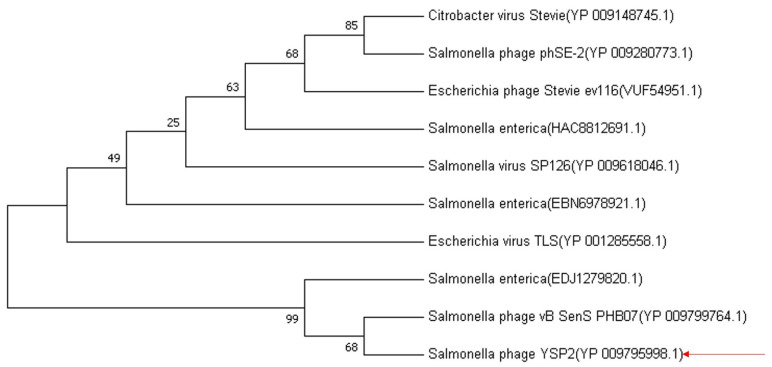
Neighbor-joining phylogenetic tree based on the amino acid fragment of LySP2 and related sequences. Bootstrap values > 50% (based on 1000 replicates) are shown at branch points. GenBank accession numbers are given in parentheses following phage names.

**Figure 2 viruses-15-00836-f002:**
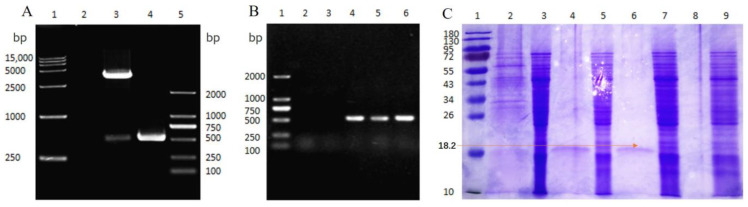
Construction and expression of LySP2. (**A**) Verification of pPICZ–LySP2 by enzyme digestion. Lanes: 1, DL 15,000 DNA marker; 2, H_2_O; 3, pPICZ–LySP2 digestion with *Xho*I and *Xba*I; 4, pPICZ–LySP2 PCR amplification product; 5, DL 2000 DNA marker. (**B**) Identification of X33–pPICZ–LySP2 recombinant yeast by PCR. Lanes: 1, DL 2000 DNA marker; 2–3, negative control; 4–6, pPICZ–LySP2 PCR amplification product. (**C**) Fermentation products analyzed by SDS-PAGE. Lanes: 1, protein ladder; 2, fermentation products of X33–pPICZ after 4 days; 3, X33–pPICZ yeast cells after 4 days of fermentation; 4, 6, and 8, fermentation products of X33–pPICZ–LySP2 after 4, 3, and 2 days; 5, 7, and 9, X33–pPICZ–LySP2 yeast cells after 4, 3, and 2 days of fermentation.

**Figure 3 viruses-15-00836-f003:**
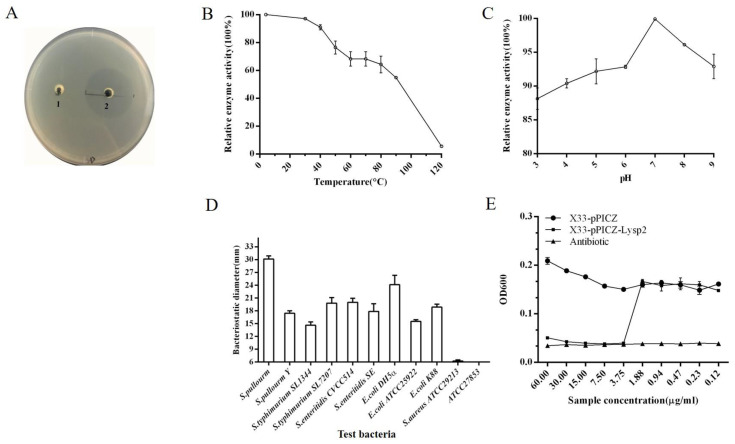
General biological characteristics of LySP2. (**A**) Activity of LySP2; add pPICZ to hole 1 and LySP2 to hole 2. (**B**) Effect of temperature on LySP2 activity. (**C**) Effect of pH on LySP2 activity. (**D**) Antibacterial activity of LySP2 against test bacteria. (**E**) Minimal inhibitory concentration of LySP2. Values are the mean ± SD of three determinations.

**Figure 4 viruses-15-00836-f004:**
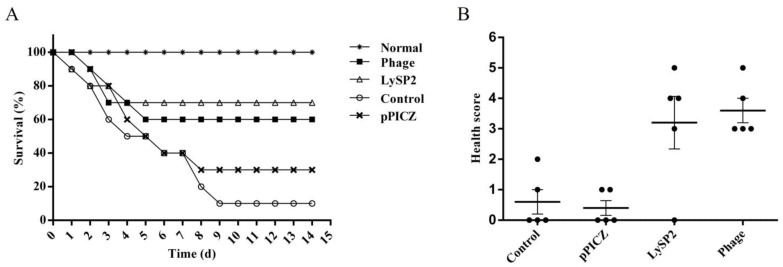
LySP2 improves the survival and health of infected chicks. (**A**) Chick survival rate from *Salmonella* pullorum infection in different groups. (**B**) Health status of *Salmonella* pullorum-infected chicks in different groups. Values are the mean ± SD of three determinations.

**Figure 5 viruses-15-00836-f005:**
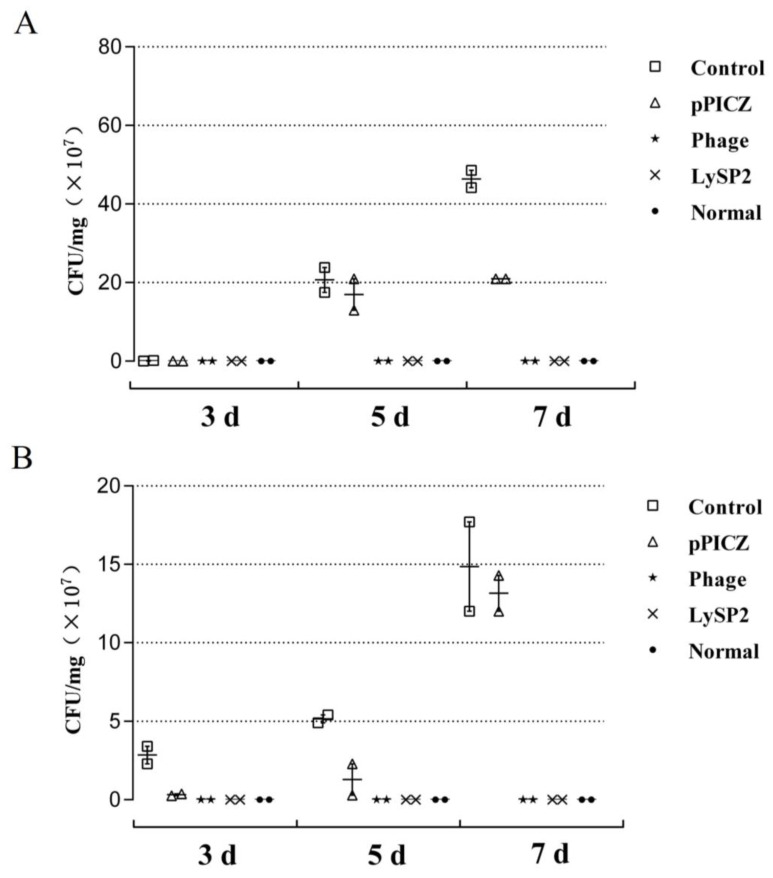
Bacterial counts. Liver (**A**) and intestinal (**B**) *Salmonella* count. Values are the mean ± SD of three determinations.

**Figure 6 viruses-15-00836-f006:**
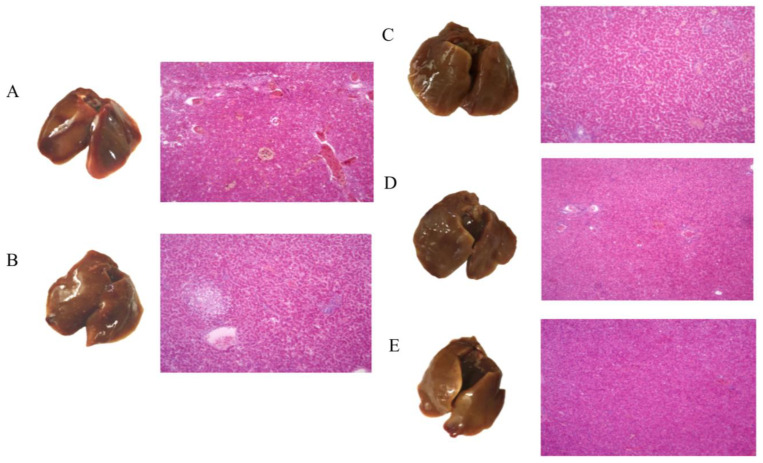
Histopathologic H&E assessment of liver and tissue samples (100×). (**A**) Control group, (**B**) pPICZ group, (**C**) Phage group, (**D**) LySP2 group, (**E**) Normal group.

**Figure 7 viruses-15-00836-f007:**
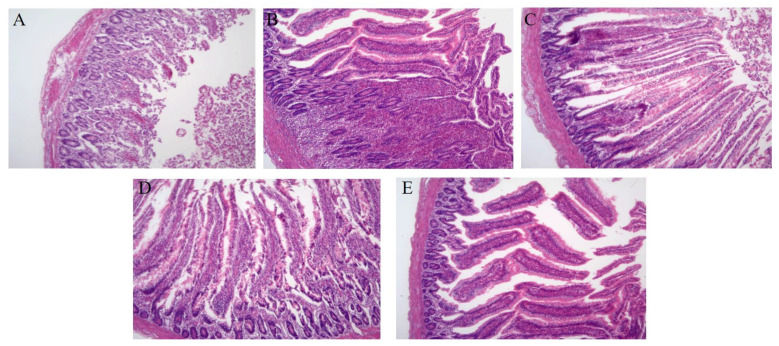
Intestinal H&E staining (100×). (**A**) Control group, (**B**) pPICZ group, (**C**) Phage group, (**D**) LySP2 group, (**E**) Normal group.

**Table 1 viruses-15-00836-t001:** Source of Bacterial Strains.

Organism	Strain	Source
*Salmonella* pullorum	SP	1
*Salmonella* pullorum	Y	1
*Salmonella* typhimurium	SL1344	2
*Salmonella* typhimurium	SL7207	2
*Escherichia coli*	K88	2
*Escherichia coli*	DH5α	3
*Salmonella* enteritidis	CVCC514	2
*Salmonella* enteritidis	SE	2
*Escherichia coli*	ATCC 25922	4
*Staphylococcus aureus*	ATCC 29212	4
*Pseudomonas aeruginosa*	ATCC 27853	4

Note: 1, isolated from the farm; 2, purchased from China Institute of Veterinary Drug Control (Beijing, China); 3, purchased from TIANGEN Biotech (Beijing) Co., Ltd. (Beijing, China); 4, purchased from the American Type Culture Collection (Gaithersburg, MD, USA).

**Table 2 viruses-15-00836-t002:** Chick grouping and processing.

Group	Challenge	Challenge Dose	Treatment
LySP2	10^6^ CFU/mL SP	0.5 mL	5 mL 60 μg/mL LySP2
Phage	10^6^ CFU/mL SP	0.5 mL	5 mL 10^10^ PFU/mL phage YSP2
pPICZ	10^6^ CFU/mL SP	0.5 mL	5 mL Empty yeast fermentation supernatant pPICZ
Control	10^6^ CFU/mL SP	0.5 mL	Drinking water
Normal	Sterile saline	0.5 mL	Drinking water

Note: Treatment: Add endolysin, phage, and empty yeast fermentation supernatant to the drinking water of the corresponding group. Refresh drinking water twice a day and stop chicks from drinking for 2 h before changing drinking water to ensure adequate medicine intake. Except for the control group, chicks in the other groups were infected with 10^6^ CFU/mL SP orally. The concentration of oral bacterial solution was determined according to the median lethal dose of *Salmonella* pullorum measured by previous laboratory work. Treatment: Except for the SP group, other groups were treated with different drugs.

**Table 3 viruses-15-00836-t003:** Endotoxin detection.

Group	LySP2	Phage	pPICZ	Control	Normal
Endotoxin concentration	0.083 ± 0.001 ^c^	0.076 ± 0.001 ^b^	0.082 ± 0.004 ^c^	0.085 ± 0.001 ^c^	0.062 ± 0.000 ^a^

Note: Same letter (*p* > 0.05), a different letter (*p* < 0.05), endotoxin: unit/mL.

**Table 4 viruses-15-00836-t004:** Chick body temperature (°C) after treatment.

Time after Treatment	LySP2	Phage	pPICZ	Control	Normal
5 days	37.45 ± 0.41	37.65 ± 0.27	37.18 ± 0.63	37.13 ± 0.25	37.25 ± 0.69
7 days	36.45 ± 0.34 ^b^	37.60 ± 0.37 ^a^	36.50 ± 0.27 ^b^	36.58 ± 0.35 ^b^	37.65 ± 0.17 ^a^
9 days	37.78 ± 0.31 ^a^	37.73 ± 0.41 ^a^	37.38 ± 0.39 ^a^	37.60 ± 0.28 ^a^	38.10 ± 0.33 ^b^

Note: Same letter (*p* > 0.05), a different letter (*p* < 0.05).

## Data Availability

Data is available on request.

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
