# Peer review of "The Broad-Spectrum Endolysin LySP2 Improves Chick Survival after Salmonella Pullorum Infection"

_viruses, 2023, doi:10.3390/v15040836_

Round 1

Reviewer 1 Report

In this manuscript, the authors analyze the function of the Gram-negative bacteriophage endolysin LySP2 in the treatment of Salmonella pullorum disease. They want to try to show that LySP2 reduces liver and intestinal lesions and improves survival in chicks infected with Salmonella. However, the results are not convincing due to missing the key purification of LySP2 and allow for other possible interpretations than the one proposed by the authors. I have the following major comments:

(1) The construction and expression of LySP2 were detected only by PCR and SDS-PAGE. In Fig. 2C, in addition to the target protein, there are numerous none-specifically expressed proteins. The expressed protein should be further purified. Moreover, SDS-PAGE only conjectured that a protein with a molecular weight of 18.2KD was expressed, but could not suggest whether this protein was LySP2. The authenticity of LySP2 expression should be directly tested by western blot assay.

(2) In Fig. 4A, safety tests of LySP2 in chicks should be performed before testing therapeutic effect of LySP2 Salmonella pullorum-infected chick. By the way, "Control group" should be changed to "Normal group" and "SP group" should be the "control group". Why did the author choose oral treatment over other methods? Are there any references for determining the dose for LySP2 in-vivo experiments and scoring criteria for the health status of chicks?

(3) Figure 6 is too blurred to see the specific lesion features.

(4) Currently, the most effective method for the prevention and control of Salmonella pullorum is purification. Authors should look for perspectives on prevention methods for LySP2 against Salmonella pullorum, rather than treatment.

Author Response

Response to Reviewer 1 Comments

Point 1:

In this manuscript, the authors analyze the function of the Gram-negative bacteriophage endolysin LySP2 in the treatment of Salmonella pullorum disease. They want to try to show that LySP2 reduces liver and intestinal lesions and improves survival in chicks infected with Salmonella. However, the results are not convincing due to missing the key purification of LySP2 and allow for other possible interpretations than the one proposed by the authors. I have the following major comments:

(1) The construction and expression of LySP2 were detected only by PCR and SDS-PAGE. In Fig. 2C, in addition to the target protein, there are numerous none-specifically expressed proteins. The expressed protein should be further purified. Moreover, SDS-PAGE only conjectured that a protein with a molecular weight of 18.2KD was expressed, but could not suggest whether this protein was LySP2. The authenticity of LySP2 expression should be directly tested by western blot assay.

Response 1: Please provide your response for Point 1. (in red)

Thank you for your valuable comments on my manuscript.

Regarding the fact that the protein is not purified, I have the following explanation,

  1. We did not add identification tags to obtain natural proteins when designing expression strategies. Because LySP2 is a kind of antibacterial protein. We pay more attention to the phenotype of the protein function than the verification at the molecular level. In addition to identifying protein expression by SDS-PAGE, it can also be confirmed by its bacteriostatic test that its expression and its lytic activity. We constructed an empty plasmid expression strain X33-pPICZ and performed the same operation as a control, and there were significant differences in the experimental results in subsequent experiments. Figure 3A shows that LySP2 had antibacterial activity in hole 2 but no activity of pPICZ in hole 1.
  2. Our initial design for LySP2 is to apply it directly after expression. However, if it is purified or has passed the western blot test, it is inevitable to add the corresponding label. In this case, removing the label before direct use will increase the related cost in the later industrial production process.
  3. Therefore, we chose the method of filtration for simple purification. (2.5. Induced fermentation and optimization of recombinant P. pastoris X33-pPICZ-LySP2)

Point 2:

(2) In Fig. 4A, safety tests of LySP2 in chicks should be performed before testing therapeutic effect of LySP2 Salmonella pullorum-infected chick. By the way, "Control group" should be changed to "Normal group" and "SP group" should be the "control group". Why did the author choose oral treatment over other methods? Are there any references for determining the dose for LySP2 in-vivo experiments and scoring criteria for the health status of chicks?

Response 2: Please provide your response for Point 2. (in red)

Thank you for your reminder about the safety test. We have done it. The protein solution was coated on the LB medium as a sterility test, and the safety test was performed in chicks alone, but it was removed from the text in the later revision. Now, we have put the experimental results in the supplementary material(lane:70-82).

I have modified the nomenclature of the groupings in the manuscript according to your suggestion, "Control group" was changed to "Normal group" and "SP group" was changed to "control group".

For the oral method, I have the following explanation: one is the infection route of Salmonella, which is transmitted through fecal-oral transmission, so we use oral methods for infection, so the corresponding oral treatment is also used to achieve rapid drug delivery to the infection site; the second is that oral treatment is the simplest and most effective way in chicken farms, that is, the drug enters the body with drinking water to play a role and can minimize the stress response of animals. The experimental doses were determined based on the MIC of the protein; there were references about the chick health score criteria, and we have included references in the revised manuscript. (lane:160,173, DOI: 10.1637/9604-112910-ResNote.1).

Point 3:

(3) Figure 6 is too blurred to see the specific lesion features.

Response 3: Please provide your response for Point 3. (in red)

Thanks for reminding us. We have replaced the clearer pictures in the revised manuscript (lane 332).

Point 4:

(4) Currently, the most effective method for the prevention and control of Salmonella pullorum is purification. Authors should look for perspectives on prevention methods for LySP2 against Salmonella pullorum, rather than treatment.

Response 4: Please provide your response for Point 4. (in red)

Thank you for your valuable advice. In this study, we used therapeutic experiments to confirm the effectiveness of LySP2 against Salmonella pullorum. We will actively use LySP2 to eliminate Salmonella in farms in the later practice, remove pathogenic Salmonella, and prevent the occurrence of Salmonella in chicken farms.

Reviewer 2 Report

In the methodology section 2.1. the author need to elaborate the isolation procedure for Salmonella and phage ysp2

Author Response

Response to Reviewer 2 Comments

Point 1:

In the methodology section 2.1. the author need to elaborate the isolation procedure for Salmonella and phage ysp2

Response 1: Please provide your response for Point 1. (in red)

Thank you for your suggestion. The Salmonella used in our experiment was isolated from chicken farms with diseased chickens. Using this Salmonella isolate as a host, we isolated phage YSP2 from the wastewater system in Changchun City.

I have included a description of the relevant isolation procedure and relevant references in the revised manuscript(lane:60, DOI: 10.1007/s11262-018-1549-0).

Reviewer 3 Report

The emergence and spread of antibiotic-resistant bacteria is a serious global problem affecting both public health as well as agriculture and food industry. For example, Salmonella enterica subsp. enterica serovar Pullorum is an important avian pathogen causing substantial losses in poultry industry. In the present submission, the Authors set an ambitious goal to use a 164-aa endolysin of phage YSP2 of Salmonella enterica subsp. enterica serovar Pullorum as an antimicrobial agent to combat pullorum disease, which is a cause of high mortality in young chickens and turkeys.

The enzyme has been overproduced in Pichia pastoris and purified.

Specific comments:

Abstract and Introduction.

1.       The title. Considering the antibacterial spectrum (see Fig. 3) the enzyme can hardly be considered a broad-spectrum lysin.

2.       The authors state that endolysins are peptidoglycan hydrolases (lane 15, lane 43, lane 366). While this is true for most lytic enzymes; however, there are lytic enzymes exemplified by lytic transglycosylases (e.g., phage lambda endolysin) that perform non-hydrolytic cleavage of the glycan strand of bacterial murein with the formation of the non-reducing N-acetyl 1,6-anhydromuramic acid, and N-acetylglucosamine [see Eur. J. Biochem 53 (1975) 47-54; J. Bacteriol. 124 (1975) 1067-1076]. Not all lytic enzymes hydrolyze the murein layer. These statements need correction.

3.       Through the text and figure legends, stick to the official Salmonella nomenclature (J. Clin. Microbiol. vol. 38 (2000) 2465).

Materials and methods. Please be specific in describing the methods used. Readers should be provided with all the necessary information.

1.       Bacterial strains and their sources should be listed.

2.       Amino acid sequence analysis of LySP2 (lane 62). Please be more specific in this section. There is a lack of accession number to the LySP2 protein (YP_009795998) and YSP2 phage (NC_047898).

3.       "Protein expression" should be replaced by "protein production" (lane 107). Genes are expressed while proteins are produced. Please introduce corrections through the text. The part devoted to protein purification is absent. Please explain in detail what kind of purification procedure was used and its rationale. Also, in the results section, authors should include information on the yield of recombinant protein purification.

Results.

4.       Section 3.1 The phage YSP2 lytic module (tandem consisting of endolysin and holin) should be described in detail. Comparative analysis should lead to a conclusion about the putative specificity of LySP2. Is this enzyme a muramidase? Please, provide a clear answer.

5.       Fig. 5. Please, correct the typos in the serovar name.

Discussion.

6.       The discussion section is, in some parts, superficial and misleading and contains untrue statements, which need correction. Please, rewrite the first paragraph of the discussion; holins do not provide phage entry to the cell at the initial stage of infection. Holins are involved in the final stage of the phage life cycle, leading to cell lysis to release phage progeny. Please read the papers from Ry Young's lab carefully.
In the case of your enzyme, you are dealing with lysis from without. To reach the murein layer, the lysin needs to pass through the outer membrane, in the case of Gram-negative bacteria. It would be appropriate to overview recent findings regarding factors facilitating outer membrane permeabilization. In this respect, an interesting discovery was made recently regarding a viral homolog of PGRP proteins (doi:10.1038/s41598-018-37417-6). It was proved for Thermus phage Ts2631 endolysin that positively charged 20-aa N-terminal extension is critical for exerting antibacterial activity. It allows the enzyme to pass through the outer membrane and interact with murein. The same research group showed the effectiveness of the Ts2631 endolysin in combating multidrug-resistant gram-negative pathogens of Acinetobacter baumannii and Pseudomonas aeruginosa (doi:10.3390/v11070657). Another example of this kind of protein is LysC of Clostridium intestinale which uses N-terminal extension to form nanopores in the cytoplasmic membrane (doi:10.3390/ijms21144894) and is capable of killing bacteria by destabilization of bacterial cell membrane through electrostatic interactions.

Author Response

Response to Reviewer 3 Comments

Point 1:

The emergence and spread of antibiotic-resistant bacteria is a serious global problem affecting both public health as well as agriculture and food industry. For example, Salmonella enterica subsp. Enterica serovar Pullorum is an important avian pathogen causing substantial losses in poultry industry. In the present submission, the Authors set an ambitious goal to use a 164-aa endolysin of phage YSP2 of Salmonella enterica subsp. Enterica serovar Pullorum as an antimicrobial agent to combat pullorum disease, which is a cause of high mortality in young chickens and turkeys.

The enzyme has been overproduced in Pichia pastoris and purified.

Specific comments:

Abstract and Introduction.

  1. The title. Considering the antibacterial spectrum (see Fig. 3) the enzyme can hardly be considered a broad-spectrum lysin.

Response 1: Please provide your response for Point 1. (in red)

Thank you for your suggestion. The "Broad-Spectrum" mentioned in the article is relative, it is broad-spectrum relative to phages because phages are very specific, but his endolysin can lyse other strains, so we call it broad-spectrum endolysin.

Point 2:

  1. The authors state that endolysins are peptidoglycan hydrolases (lane 15, lane 43, lane 366). While this is true for most lytic enzymes; however, there are lytic enzymes exemplified by lytic transglycosylases (e.g., phage lambda endolysin) that perform non-hydrolytic cleavage of the glycan strand of bacterial murein with the formation of the non-reducing N-acetyl 1,6-anhydromuramic acid, and N-acetylglucosamine [see Eur. J. Biochem 53 (1975) 47-54; J. Bacteriol. 124 (1975) 1067-1076]. Not all lytic enzymes hydrolyze the murein layer. These statements need correction.

Response 2: Please provide your response for Point 2. (in red)

Thank you for your valuable and professional suggestions and references, which allowed us to correct manuscript errors and broadened my knowledge points. According to your guidance, I have made revisions in the revised manuscript (For the description of endolysin, I have replaced it with " Most of the endolysins are hydrolytic enzymes produced by bacteriophages to cleave the host's cell wall during the final stage of the lytic cycle "lane:15; Based on your suggestion, I have added descriptions of enzymes with other cleavage mechanisms in the "Introduction" section, lane: 47-50).

Point 3:

  1. Through the text and figure legends, stick to the official Salmonella nomenclature (J. Clin. Microbiol. vol. 38 (2000) 2465).

Response 3: Please provide your response for Point 3. (in red)

Thank you for reminding me. I have corrected the revised manuscript, as requested by the "Instructions for Authors" of the viruses (modification details can be seen in table 1, lane:137).

Point 4:

Materials and methods. Please be specific in describing the methods used. Readers should be provided with all the necessary information.

  1. Bacterial strains and their sources should be listed.

Response 4: Please provide your response for Point 4. (in red)

Thanks for your suggestion. We should describe the strains’ source clearly; I have added a table in the "Determination of the endolysin range of LySP2" section of the revised manuscript to describe the strains mentioned in the text.

Point 5:

  1. Amino acid sequence analysis of LySP2 (lane 62). Please be more specific in this section. There is a lack of accession number to the LySP2 protein (YP_009795998) and YSP2 phage (NC_047898).

Response 5: Please provide your response for Point 5. (in red)

Thanks for the suggestion. I have added the accession numbers to the revised manuscript (lane:58,67).

Point 6:

  1. "Protein expression" should be replaced by "protein production" (lane 107). Genes are expressed while proteins are produced. Please introduce corrections through the text. The part devoted to protein purification is absent. Please explain in detail what kind of purification procedure was used and its rationale. Also, in the results section, authors should include information on the yield of recombinant protein purification.

Response 6: Please provide your response for Point 6. (in red)

Thank you for your suggestions and revisions. I have revised the manuscript according to your suggestions(lane:108,112,408);

The size of our protein is 18.2KDa, so we added the supernatant expressing the protein to a 50-kDa molecular weight cut-off ultrafiltration device, and the filtered supernatant in the centrifugal tube was collected. The process was repeated with a 3-kDa molecular weight cut-off ultrafiltration device with centrifugation at 2000×g for 15 min. We performed simple purification and concentration by filtration.

I have the following explanation regarding the fact that the protein is not purified.

  1. We did not add identification tags to obtain natural proteins when designing expression strategies. Because LySP2 is a kind of antibacterial protein. We pay more attention to the phenotype of the protein function than the verification at the molecular level. In addition to identifying protein expression by SDS-PAGE, it can also be confirmed by its bacteriostatic test that its expression and its lytic activity. We constructed an empty plasmid expression strain X33-pPICZ and performed the same operation as a control, and there were significant differences in the experimental results in subsequent experiments. Figure 3A shows that LySP2 had antibacterial activity in hole 2 but no activity of pPICZ in hole 1.
  2. Our initial design for LySP2 is to apply it directly after expression. However, if it is purified or has passed the western blot test, it is inevitable to add the corresponding label. Removing the label before direct use will increase the related cost in the later industrial production process.
  3. Therefore, we chose the method of filtration for simple purification. (2.5. Induced fermentation and optimization of recombinant P. pastoris X33-pPICZ-LySP2)

Point 7:

Results.

  1. Section 3.1 The phage YSP2 lytic module (tandem consisting of endolysin and holin) should be described in detail. Comparative analysis should lead to a conclusion about the putative specificity of LySP2. Is this enzyme a muramidase? Please, provide a clear answer.

Response 7: Please provide your response for Point 7. (in red)

Thank you for your suggestions and revisions.

According to our bioinformatics comparison, analysis, and experimental results, LySP2 is not muramidase; it is endolysins, hydrolases produced by phages to cleave the host’s cell wall at the final stage of the lytic cycle(lane229-230).

Point 8:

  1. Fig. 5. Please, correct the typos in the serovar name.

Response 8: Please provide your response for Point 8. (in red)

Thank you for your suggestions and revisions. I have corrected it.

Point 9:

Discussion.

  1. The discussion section is, in some parts, superficial and misleading and contains untrue statements, which need correction. Please, rewrite the first paragraph of the discussion; holins do not provide phage entry to the cell at the initial stage of infection. Holins are involved in the final stage of the phage life cycle, leading to cell lysis to release phage progeny. Please read the papers from Ry Young's lab carefully.

In the case of your enzyme, you are dealing with lysis from without. To reach the murein layer, the lysin needs to pass through the outer membrane, in the case of Gram-negative bacteria. It would be appropriate to overview recent findings regarding factors facilitating outer membrane permeabilization. In this respect, an interesting discovery was made recently regarding a viral homolog of PGRP proteins (doi:10.1038/s41598-018-37417-6). It was proved for Thermus phage Ts2631 endolysin that positively charged 20-aa N-terminal extension is critical for exerting antibacterial activity. It allows the enzyme to pass through the outer membrane and interact with murein. The same research group showed the effectiveness of the Ts2631 endolysin in combating multidrug-resistant gram-negative pathogens of Acinetobacter baumannii and Pseudomonas aeruginosa (doi:10.3390/v11070657). Another example of this kind of protein is LysC of Clostridium intestinale which uses N-terminal extension to form nanopores in the cytoplasmic membrane (doi:10.3390/ijms21144894) and is capable of killing bacteria by destabilization of bacterial cell membrane through electrostatic interactions.

Response 9: Please provide your response for Point 9. (in red)

Thank you for your professional advice and for helping us correct our mistakes. Regarding holins, we should give a more precise definition. Thank you for the references in other studies on the cleavage mechanism of endolysins, which helped us enrich the discussion section. I have added relevant content in the revised manuscript(lane:369-374).

Reviewer 4 Report

Dear Authors,

Appreciating your efforts. Enclosed you can find my comments and suggestions that might help improving the manuscript.

In general:

1- Try to unify the format and state the full text of the abbreviations in the first mention.

2- What about the economical impact of your valuable work? Do you assess it?

3- Methodology require references. 

4- Statistical analysis needs more details about the experimental design and the test to differentiate among means (mor than 2 means) (for example: Duncan or Tukey).

4- Update references for 2022 and/or 2023.

5- Improve your discussion.

Author Response

Response to Reviewer 4 Comments

Point 1:

In general:

1- Try to unify the format and state the full text of the abbreviations in the first mention.

Response 1: Please provide your response for Point 1. (in red)

Thank you for your suggestion. I have made corrections in the revised manuscript (lane 104).

Point 2:

2- What about the economical impact of your valuable work? Do you assess it?

Response 2: Please provide your response for Point 2. (in red)

Thank you for your positive evaluation of my work. Our work is still in the primary research stage. Although the health status of infected chicks has been significantly improved in animal experiments, it is still far from clinical use. However, our research group has previous experience with clinically used antimicrobial peptides, and we can apply our current work to the clinic as soon as possible.

Point 3:

3- Methodology require references.

Response 3: Please provide your response for Point 3. (in red)

Thanks for your suggestion. I will add references to the revised manuscript (lane 59 DOI: 10.1007/s11262-018-1549-0; lane 162 DOI: 10.3382/ps.2008-00410; lane 175 DOI: 10.1637/9604-112910-ResNote.1 ).

Point 4:

4- Statistical analysis needs more details about the experimental design and the test to differentiate among means (more than 2 means) (for example: Duncan or Tukey).

Response 4: Please provide your response for Point 4. (in red)

Thanks for your suggestion, and we got three sets of experimental data and obtained the experimental results through the Tukey test.

Point 5:

Update references for 2022 and/or 2023.

Response 5: Please provide your response for Point 5. (in red)

Thanks for your suggestion. I’ve added an up-to-date reference in the Introduction and Discussion sections in the revised manuscript (DOI: 10.3390/foods12020411; DOI: 10.1007/s00203-023-03434-x; DOI: 10.3390/v15020520; DOI: 10.3390/v14020167).

Point 6:

- Improve your discussion.

Response 6: Please provide your response for Point 6. (in red)

Thank you for your valuable suggestions. The Discussion section does need to be seriously revised and enriched. Other reviewers also gave the same suggestion, so I actively revised and enhanced the discussion section in the revised manuscript (lane:360-363, I improved the description of holin; lane:371-376, Multiple enzymes with other cleavage mechanisms are listed; lane:394-395; lane:429-448;).

Finally, I sincerely thank you for your revision and annotation of my manuscript word by word. You are the most serious, responsible, and professional reviewer ever. Thank you for every correction and suggestion, which is of great help to this manuscript and every future writing of mine.

Round 2

Reviewer 4 Report

The section of statistical analysis requires improvement. T-test valid for 2 means, as you mention in the reply to reviewer, Tukey's test (Honest significant difference) was applied, however it is NOT mentioned in the manuscript. Also, state clearly the experimental design of the research work. 

Author Response

Response to Reviewer 4 Comments

Round 2

Point 1: The section of statistical analysis requires improvement. T-test valid for 2 means, as you mention in the reply to reviewer, Tukey's test (Honest significant difference) was applied, however it is NOT mentioned in the manuscript.

Response 1: Please provide your response for Point 1. (in red)

Thank you again for your advice and reminder, I have added the relevant description in the "Statistical analysis" section.

The "Statistical analysis" section has been revised: The data are expressed as mean±SD. All experiments were independently per-formed three times. T-tests were applied to assess the difference when two groups were compared (where P<0.05 indicates significant differences and P<0.01 shows extremely significant differences) . Tukey test was applied when examining changes in endotoxin and body temperature in different groups (where P<0.05 indicates significant difference-ences). Statistical analysis was performed with GraphPad Software (GraphPad Software Inc., San Diego, CA, USA) .(lane:208-214)

Point 2: Also, state clearly the experimental design of the research work. 

Response 2:Please provide your response for Point 2. (in red)

Thanks for the suggestion, I have clarified the experimental design in the "Introduction" section.

I provide a summary of the experimental design at the end of the Discussion section, described as follows " In this study, the endolysin expressed by Pichia pastoris showed high lytic activity when used directly. The experimental design of this work can be roughly described as follows: The natural endolysin with antibacterial activity was produced by a eukaryotic expression system, the endolysin was secreted and expressed in the culture supernatant, then characterize its biological properties and antibacterial tests in vitro, direct oral use in animal experiments to verify the therapeutic effect of endolysin on infected animals.”(lane:51-57)
